# Neuroprotection with the Cannabidiol Quinone Derivative VCE-004.8 (EHP-101) against 6-Hydroxydopamine in Cell and Murine Models of Parkinson’s Disease

**DOI:** 10.3390/molecules26113245

**Published:** 2021-05-28

**Authors:** Sonia Burgaz, Concepción García, María Gómez-Cañas, Alain Rolland, Eduardo Muñoz, Javier Fernández-Ruiz

**Affiliations:** 1Department of Biochemistry and Molecular Biology, Faculty of Medicine, Institute on Neurochemistry Research, Complutense University, 28040 Madrid, Spain; soniabur@ucm.es (S.B.); conchig@med.ucm.es (C.G.); mgc@med.ucm.es (M.G.-C.); 2Centro de Investigación Biomédica en Red de Enfermedades Neurodegenerativas (CIBERNED), 28031 Madrid, Spain; 3Instituto Ramón y Cajal de Investigación Sanitaria (IRYCIS), 28034 Madrid, Spain; 4Emerald Health Pharmaceuticals, San Diego, CA 92121, USA; arolland@emeraldpharma.life (A.R.); fi1muble@uco.es (E.M.); 5Instituto Maimónides de Investigación Biomédica de Córdoba (IMIBIC), 14004 Córdoba, Spain; 6Department of Cellular Biology, Physiology and Immunology, University of Córdoba, 14004 Córdoba, Spain; 7Hospital Universitario Reina Sofía, 14004 Córdoba, Spain

**Keywords:** cannabinoids, VCE-004.8, EHP-101, PPAR-γ receptors, CB_2_ receptors, mitochondrial dysfunction, 6-hydroxydopamine, Parkinson’s disease

## Abstract

The 3-hydroxyquinone derivative of the non-psychotrophic phytocannabinoid cannabigerol, so-called VCE-003.2, and some other derivatives have been recently investigated for neuroprotective properties in experimental models of Parkinson’s disease (PD) in mice. The pharmacological effects in those models were related to the activity on the peroxisome proliferator-activated receptor-γ (PPAR-γ) and possibly other pathways. In the present study, we investigated VCE-004.8 (formulated as EHP-101 for oral administration), the 3-hydroxyquinone derivative of cannabidiol (CBD), with agonist activity at the cannabinoid receptor type-2 (CB_2_) receptor in addition to its activity at the PPAR-γ receptor. Studies were conducted in both in vivo (lesioned-mice) and in vitro (SH-SY5Y cells) models using the classic parkinsonian neurotoxin 6-hydroxydopamine (6-OHDA). Our data confirmed that the treatment with VCE-004.8 partially reduced the loss of tyrosine hydroxylase (TH)-positive neurons measured in the substantia nigra of 6-OHDA-lesioned mice, in parallel with an almost complete reversal of the astroglial (GFAP) and microglial (CD68) reactivity occurring in this structure. Such neuroprotective effects attenuated the motor deficiencies shown by 6-OHDA-lesioned mice in the cylinder rearing test, but not in the pole test. Next, we explored the mechanism involved in the beneficial effect of VCE-004.8 in vivo, by analyzing cell survival in cultured SH-SY5Y cells exposed to 6-OHDA. We found an important cytoprotective effect of VCE-004.8 at a concentration of 10 µM, which was completely reversed by the addition of antagonists, T0070907 and SR144528, aimed at blocking PPAR-γ and CB_2_ receptors, respectively. The treatment with T0070907 alone only caused a partial reversal, whereas SR144528 alone had no effect, indicating a major contribution of PPAR-γ receptors in the cytoprotective effect of VCE-004.8 at 10 µM. In summary, our data confirmed the neuroprotective potential of VCE-004.8 in 6-OHDA-lesioned mice, and in vitro studies confirmed a greater relevance for PPAR-γ receptors rather than CB_2_ receptors in these effects.

## 1. Introduction

Phytocannabinoids, the active constituents of Cannabis plant, and also endocannabinoids and synthetic cannabinoids have been proposed as promising neuroprotective agents, a property derived from their pleiotropism and ability to activate numerous cytoprotective targets within the endocannabinoid system, but also outside this signaling system (reviewed in [1]). Such neuroprotective potential has been preclinically investigated in accidental brain damage (e.g., stroke, brain trauma, spinal injury) and, in particular, in chronic progressive disorders (e.g., Alzheimer’s disease, amyotrophic lateral sclerosis, Huntington’s disease, and others) [2]. This also includes Parkinson’s disease (PD), which is one of the disorders that has attracted to date most of the research with cannabinoids, aimed at exploring neuroprotective therapies to delay or arrest disease progression and also alleviate specific parkinsonian symptoms (reviewed in [2,3,4,5]). Some of these studies have concentrated on compounds targeting the cannabinoid type-1 (CB_1_) receptor, demonstrating neuroprotective properties in some experimental models of PD [6,7]. However, most of the experimental evidence obtained with this receptor concentrated on the capability to reduce specific parkinsonian signs such as bradykinesia and immobility [8,9,10], tremor [11], and/or L-dihydroxyphenylalanine (L-DOPA)-induced dyskinesia [12]. In contrast, the neuroprotective potential of cannabinoids in PD was initially associated with compounds, such as cannabidiol, having an antioxidant profile exerted by cannabinoid receptor-independent effects [13] or through modulating nuclear factor erythroid 2-related factor 2 (Nrf-2) signaling [14]. Later on, strong neuroprotective properties in PD were found for those cannabinoids active against inflammation and glial reactivity, whose effects are exerted through the activation of the cannabinoid type-2 (CB_2_) receptor [15,16,17,18], but also targeting the peroxisome proliferator-activated receptor-γ (PPAR-γ) [19,20] and the G protein-coupled receptor 55 (GPR55) [21].

These studies have identified several promising cannabinoid compounds to generate a cannabinoid-based therapy for specific symptoms and, in particular, for disease progression in patients affected by PD. One such compound is the phytocannabinoid Δ^9^-tetrahydrocannabivarin (Δ^9^-THCV), which is active in alleviating motor inhibition [16] or delaying L-DOPA-induced dyskinesia [22] by its CB_1_ receptor antagonist activity. It also displays an important anti-inflammatory and neuroprotective profile in 6-hydroxydopamine (6-OHDA)- and lipopolysaccharide (LPS)-lesioned mice exerted through multiple mechanisms, in particular by activating the CB_2_ receptor [16,18]. A second interesting compound is the non-thiophilic cannabigerol (CBG) quinone derivative VCE-003.2, which behaves as a PPAR-γ activator with no activity at the CB_1_/CB_2_ receptors [23]. It has been found to be active as anti-inflammatory and neuroprotectant against inflammation-driven neuronal deterioration in LPS-lesioned mice [19,20], and also in 6-OHDA-lesioned mice [24]. These effects were found to be mediated by its binding at a functional alternative site different from the canonical binding site used by glitazones in the PPAR-γ receptor, as shown in in vitro studies [19,24]. A third group of promising compounds are the cannabigerolic acid analog CBGA-Q and its salt form [24]. Similarly to VCE-003.2, both exhibited a notable effect as neuroprotective agents in 6-OHDA-lesioned mice, but the neuroprotective effects were mediated by the activation of the canonical binding site in the PPAR-γ receptor [24]. The structures of all these synthetic compounds were previously disclosed [23,24].

Another interesting compound that has not been investigated in PD yet is the cannabidiol (CBD) aminoquinone derivative, VCE-004.8 (Emerald Health Pharmaceuticals, USA; see chemical structure in Figure 1). VCE-004.8 has the ability to activate the CB_2_ receptor in addition to the PPAR-γ receptor [25], which may be of interest in PD following the results found in studies selectively activating the CB_2_ receptor [16,18]. This compound is presently under clinical investigation (formulated as EHP-101 oral solution) for autoimmune disorders (https://clinicaltrials.gov/ct2/show/NCT04166552 accessed on 15 May 2021), such as systemic sclerosis, a rare form of scleroderma, and for multiple sclerosis. In the present study, we first investigated the neuroprotective effect of VCE-004.8 (EHP-101) in 6-OHDA-lesioned mice using an oral administration. Second, we also investigated the potential targets for these effects (PPAR-γ and/or CB_2_ receptors) using cultured SH-SY5Y cells exposed to 6-OHDA.

## 2. Results

### 2.1. Neuroprotective Effects of VCE-004.8 in 6-OHDA-Lesioned Mice

Our first experiment explored the neuroprotective potential of an oral administration of VCE-004.8 at the dose of 20 mg/kg in 6-OHDA-lesioned mice. After 2 weeks of daily treatment, we first investigated the neurological status of these mice using two motor tests (pole and cylinder rearing tests). In the first test, our 6-OHDA-lesioned mice spent more time in descending the pole than control (sham) mice, but the treatment with VCE-004.8 was unable to reverse this defect (F(2,15) = 1.356, ns; Figure 2). This was not the case in relation with the hemiparesis shown in the cylinder rearing test (this is a more reliable test for testing parkinsonism signs when unilateral models are used) by 6-OHDA-lesioned mice compared to controls, which was partially recovered after the treatment with VCE-004.8 (F(2,12) = 13.40, *p* < 0.005; Figure 2).

This positive effect was concordant with a small (<40%) but significant preservation in the number of tyrosine hydroxylase (TH)-positive nigral neurons (F(2,16) = 28.42, *p* < 0.0001; Figure 3), accompanied by a much more evident reduction in the elevated glial reactivity labelled with two proteins: cluster of differentiation (Cd68) (F(2,16) = 17.07, *p* < 0.0005; Figure 4) and glial fibrillary acidic protein (GFAP) (2,16) = 16.59, *p* < 0.0005; Figure 5).

### 2.2. Effects of VCE-004.8 against 6-OHDA Insult in Cultured SH-SY5Ycells

The second part of our study consisted of a series of experiments conducted in a cell-based assay (cultured SH-SY5Y cells exposed to 6-OHDA) that is frequently used as an in vitro model of PD [26], and that, in our case, was used as a way to confirm the in vivo data and, in particular, for exploring the potential targets (PPAR-γ and/or CB_2_ receptors) involved in the beneficial effects found with this CBD derivative. First, we carried out a concentration-response experiment to determine the best VCE-004.8 concentration to increase cell survival against the 6-OHDA (200 µM) insult according to similar experiments conducted in the same cell-based model with other phytocannabinoid derivatives [24]. VCE-004.8 showed cytoprotection in a concentration-dependent manner (F(5,35) = 116.6, *p* < 0.0001; Figure 6) in the range 2–10 µM, with higher concentrations resulting in being less efficacious (20 µM) or even more toxic (40 µM) (Figure 6). Once having selected 10 µM as the most adequate concentration, we investigated the cytoprotective effect of this concentration of VCE-004.8 against 200 µM 6-OHDA, in the presence or absence of the PPAR-γ receptor inhibitor T0070907 (10 µM) to inactivate PPAR-γ receptors, the selective antagonist SR144528 (10 µM) to block CB_2_ receptors, or the combination of both compounds. Our data confirmed that 200 µM 6-OHDA reduced cell viability up to 50%, which was significantly elevated (up to 70%) with VCE-004.8 (F(7,47) = 49.0, *p* < 0.0001; Figure 6). This cytoprotective effect was completely eliminated only with the combination of T0070907 and SR144528 together (Figure 6), thus confirming that both PPAR-γ and CB_2_ receptors contribute to the effect of VCE-004.8. However, whereas the treatment with T0070907 alone caused a partial reversal, the treatment with SR144528 alone has no effect (Figure 6), thus indicating a major contribution of PPAR-γ receptors in the cytoprotective effect of VCE-004.8 at 10 µM. Both antagonists have no effect when added alone to cell cultures (Figure 6).

## 3. Discussion

The experiments included here follow previous experiments aimed at exploring the neuroprotective potential of different CBG or CBGA quinone derivatives (e.g., VCE-003.2, CBGA-Q, CBGA-Q-Salt) in experimental models of PD [19,20,24]. The novelty of this study was the evaluation of a 3-hydroxyquinone derivative of CBD, VCE-004.8, which, compared to the other phytocannabinoid derivatives investigated in PD [19,20,24], has the particularity to be also active as an agonist of the CB_2_ receptor in addition to its activity on the PPAR-γ receptor [25]. A priori, our expectation was that this hybrid activity may provide VCE-004.8 with some advantages in terms of potency and/or broad-spectrum (multitarget) properties compared to the other compounds. Our data confirmed first that VCE-004.8 was also active in the 6-OHDA model, being able to attenuate the loss in the number of TH-positive nigral neurons caused by this neurotoxin. Our data also proved that VCE-004.8 was strongly active against the inflammatory response (elevated astrogliosis labelled with GFAP and microgliosis labelled with CD68) occurring in 6-OHDA-lesioned mice, which was completely attenuated by the treatment with this CBD derivative. A priori this effect would be presumably the result of the VCE-004.8-induced capability to preserve TH-positive nigral neurons, as glial reactivity has been found to be secondary to neuronal death in this PD model [16]. Such benefits were paralleled by recovery of the motor defects typical of 6-OHDA-lesioned mice. We used first the pole test and found no effects of VCE-004.8. Next, we used the cylinder rearing test, which is a much more adequate tool for detecting motor impairment in PD models generated by unilateral lesions, and we observed that the hemiparesis shown by 6-OHDA-lesioned mice in this test was significantly attenuated by the treatment with VCE-004.8.

Therefore, these in vivo data confirm the benefits of this CBD derivative when given orally in this experimental model of PD, as also demonstrated previously for the CBG and CBGA quinone derivatives [19,20,24]. However, our data did not show additional potency for VCE-004.8 in 6-OHDA-lesioned mice, despite its hybrid activity (CB_2_ and PPAR-γ receptors) [25]. This could be related to the fact that, whereas PPAR-γ receptors are elevated in lesioned areas in 6-OHDA-treated mice [24], CB_2_ receptors remain unaltered [16], which may limit the contribution of this last receptor. An alternative explanation may be that the combination of CB_2_/PPAR-γ activities provided by VCE-004.8 may be reflected exclusively in potentiating its anti-inflammatory activity, a fact that may be concluded from our data in GFAP and CD68 immunostaining, but without representing a greater improvement of its neuroprotective effects (no greater TH immunostaining recovery compared with CBG derivatives) in the 6-OHDA model in which glial reactivity is secondary to neuronal deterioration. If this is true, it is possible that VCE-004.8 may work better in more inflammatory models of PD such as LPS- or rotenone-lesioned mice, in which it is well-known that inflammatory events play more important roles in the pathogenesis. In addition, CB_2_ receptors become up-regulated in lesioned structures in both LPS- [16,18] and rotenone-lesioned mice [27], which has been found to facilitate the anti-inflammatory and neuroprotective effect of compounds selectively activating this receptor in these models.

Lastly, as in our previous studies with different CBG/CBGA derivatives [19,20,24], we also investigated the potential mechanism(s) involved in the effects of the CBD derivative against 6-OHDA insult using an in vitro model, SH-SY5Y cells. We first explored the best concentration (10 µM) of VCE-004.8 for having a significant increase in cell viability against the 6-OHDA insult, and we then investigated the contribution of both CB_2_ and PPAR-γ receptors to these effects using selective blockade with SR144528 and T0070907, respectively. Our in vitro data confirmed the in vivo findings that PPAR-γ receptors have a major role in the effects of VCE-004.8 in the 6-OHDA model, as reversal of its cytoprotective effects was seen only when T0070907 was added, either alone (partial reversal) or combined with SR144528 (total reversal). By contrast, the use of SR144528 alone had no effect in reversing the VCE-004.8-induced cytoprotection. Therefore, our data support that both receptors are necessary to reach the maximal reversal of this effect, but with higher relevance for PPAR-γ. In addition, our data also confirmed that the site for the effect of VCE-004.8 in the PPAR-γ was the canonical site and not the alternative regulatory site in the PPAR-γ receptor [28], which we found to be the site for the action of VCE-003.2 [19,24].

## 4. Materials and Methods

### 4.1. Synthesis and Receptor Characterization of the Different Compounds Investigated

The 3-hydroxyquinone derivative of CBD, (1′R,6′R)-3-(benzylamino)-6-hydroxy-3′-methyl-4-pentyl-6′-(prop-1-en-2-yl)-[1,1′bi(cyclohexane)]-2′,3,6-triene-2,5-dione (VCE-004.8) was synthesized as described previously (del Río et al., 2016). Its pharmacodynamic profile (PPAR-γ and CB_2_ agonist) has been previously described [25] and its oral lipid formulation (EHP-101) is currently in clinical development following its benefits found for bleomycin-induced skin and lung fibrosis [29].

### 4.2. Animals and Surgical Lesions

Male C57BL/6 mice were housed in a room with a controlled photoperiod (06:00–18:00 light) and temperature (22 ± 1 °C). They had free access to standard food and water and were used at adult age (3–4-month-old; 25–30 g weight). All experiments were conducted according to European guidelines (directive 2010/63/EU) and approved by the “Comité de Experimentación Animal” of our university (ref. CEA-UCM 56/2012). For in vivo experiments, mice were anaesthetized (ketamine 40 mg/kg + xylazine 4 mg/kg, i.p.) 30 min after pretreatment with desipramine (25 mg/kg, i.p.), and then 6-OHDA free base (2 μL at a concentration of 2 μg/μL saline in 0.2% ascorbate to avoid oxidation) or saline (for control mice) were injected stereotaxically into the right striatum at a rate of 0.5 μL/min, using the following coordinates: +0.4 mm AP, ±1.8 mm ML and −3.5 mm DV, as described by Alvarez-Fischer and coworkers [30]. Once injected, the needle was left in place for 5 min before being slowly withdrawn, thus avoiding reflux and a rapid increase in intracranial pressure. Control animals were sham-operated and injected with 2 μL of saline using the same coordinates. After the application of 6-OHDA or saline, mice were subjected to pharmacological treatments as described in the following section. The lesions were generated using unilateral injection, the advantage of which is that contralateral structures serve as controls for the different analyses.

### 4.3. Pharmacological Treatments and Sampling

VCE-004.8 (20 mg/kg, according to a previous study [25]) or vehicle (Maisine CC/corn oil) were orally administered to 6-OHDA-lesioned mice. Control mice (sham-operated) were also administered with the vehicle. The first dose was administered approximately 16 h after the lesion, and the treatment was prolonged for two weeks (one dose per day, always at the same time). One day after the last administration, all animals were analyzed in the pole test and the cylinder rearing test, at the end of which animals were killed by rapid and careful decapitation and their brains were rapidly removed and fixed for one day at 4 °C in fresh 4% paraformaldehyde prepared in 0.1 M phosphate buffered-saline (PBS), pH 7.4. Samples were cryoprotected by immersion in a 30% sucrose solution for a further day, and finally stored at −80 °C for subsequent immunostaining analysis in the substantia nigra.

### 4.4. Behavioral Procedures

#### 4.4.1. Pole Test

Mice were placed head-upward on the top of a vertical rough-surfaced pole (diameter 8 mm; height 55 cm) and the time until animals descended to the floor was recorded with a maximum duration of 120 s [31]. When the mouse was not able to turn downward and instead dropped from the pole, the time was taken as 120 s (default value).

#### 4.4.2. Cylinder Rearing Test

Given that the lesion was unilateral, this test attempted to quantify the degree of forepaw (ipsilateral, contralateral, or both) preference for wall contacts after placing the mouse in a methacrylate transparent cylinder (diameter: 15.5 cm; height: 12.7 cm; [32]). Each score was made out of a 3 min trial with a minimum of 4 wall contacts.

### 4.5. Immunohistochemical Procedures

Brains were sliced in coronal sections (containing the substantia nigra) in a cryostat (30 µm thick) and collected on antifreeze solution (glycerol/ethylene glycol/PBS; 2:3:5) and stored at −20 °C until used. Sections were mounted on gelatin-coated slides, and, once adhered, washed in 0.1M potassium PBS (KPBS) at pH 7.4. Then endogenous peroxidase was blocked by 30 min incubation at room temperature in peroxidase blocking solution (Dako Cytomation, Glostrup, Denmark). After several washes with KPBS, sections were incubated overnight at room temperature with the following primary antibodies: (i) polyclonal rabbit anti-TH (Chemicon-Millipore, Temecula, CA, USA) used at 1/200; (ii) polyclonal rat anti-mouse CD68 antibody (AbD Serotec, Oxford, UK) used at 1/200; or (iii) polyclonal rabbit anti-mouse GFAP antibody (Dako Cytomation, Glostrup, Denmark) used at 1/200. Dilutions were carried out in KPBS containing 2% bovine serum albumin and 0.1% Triton X-100 (Sigma Chem., Madrid, Spain). After incubation, sections were washed in KPBS, followed by incubation with the corresponding biotinylated secondary antibody (1/200) (Vector Laboratories, Burlingame, CA, USA) for 1 h at room temperature. Avidin-biotin complex (Vector Laboratories, Burlingame, CA, USA) and 3,3′-diaminobenzidine substrate–chromogen system (Dako Cytomation, Glostrup, Denmark) were used to obtain a visible reaction product. Negative control sections were obtained using the same protocol with omission of the primary antibody. A Leica DMRB microscope and a DFC300FX camera (Leica, Wetzlar, Germany) were used for the observation and photography of the slides, respectively. For quantification of the intensity of TH, GFAP or CD68 immunostaining either in the substantia nigra (both ipsilateral and contralateral sides), we used the NIH Image Processing and Analysis software (ImageJ; NIH, Bethesda, MD, USA) using 4–5 sections, separated approximately by 200 µm, and observed with 5×–20× objectives depending on the method and the brain area under quantification. In all sections, the same area of the substantia nigra pars compacta was analyzed. Analyses were always conducted by experimenters who were blinded to all animal characteristics. Data were expressed as percentage of immunostaining intensity in the ipsilateral (lesioned) side over the contralateral (non-lesioned) side.

### 4.6. Cultures of SH-SY5Y Neuronal Cells

Cultures of SH-SY5Y neuronal cell line (kindly provided by Dr. Ana Martínez, CIB-CSIC, Madrid, Spain) were used to induce cell death with 6-OHDA and to investigate in vitro the mechanisms of cytoprotection of the different cannabinoid derivatives under study, following a procedure described previously [33]. To this end, SH-SY5Y cells were maintained in DMEM supplemented with 10% FBS, 2 mM Ultraglutamine, and 1% antibiotics (Lonza, Verviers, Belgium) and under a humidified 5% CO_2_ atmosphere at 37 °C. For cytotoxicity experiments, cells were seeded at 60,000 cells/well in 96-well plates and maintained under a humidified atmosphere (5% CO_2_) at 37 °C overnight. In a first experiment, 24 h after seeding, cells were treated with the vehicle (DMEM + 0.1% DMSO) or with four different concentrations of VCE-004.8 (2, 10, 20, and 40 μM; selected according to Del Río et al., 2016), just 60 min before being exposed to 200 µM 6-OHDA (or saline) following our previously published studies with different concentrations of 6-OHDA in these cells [24]. Cells were incubated over 24 h before the neuronal death was analyzed with the MTT assay (Panreac AppliChem., Barcelona, Spain). This experiment served to select the best VCE-004.8 concentration for the second experiment, in which cells were treated with the vehicle (DMEM + 0.1% DMSO), with the PPAR-γ antagonist T0070907 (10 µM), with the selective CB_2_ receptor antagonist SR144528 (10 µM), or with both, followed, 30 min later, by a new treatment with VCE-004.8 (10 µM) or vehicle (DMEM + 0.1% DMSO), just 60 min before being exposed to 200 µM 6-OHDA (or saline). Cells were again incubated over 24 h before the neuronal death was analyzed with the MTT assay (Panreac AppliChem., Barcelona, Spain). In all cases, the data of cell viability were normalized in relation to the corresponding control group (cells exposed to vehicles for 6- OHDA and compounds).

### 4.7. Data Analysis

Data were normally distributed (tested with the Shapiro–Wilk normality test) and were assessed by one-way analysis of variance followed by the Bonferroni test, using GraphPad Prism^®^ software (version 5.01; GraphPad Software Inc., San Diego, CA, USA).

## 5. Conclusions

In summary, our data confirmed the neuroprotective potential of VCE-004.8 in 6-OHDA-lesioned mice, which adds to information on previously investigated cannabinoid derivatives. In vitro studies confirmed a greater relevance for PPAR-γ receptors rather than CB_2_ receptors in these effects. With this study as a whole followed by additional preclinical studies in other experimental models of PD, we expect that all these data will generate further interest in cannabinoid derivatives targeting CB_2_ and PPAR-γ receptors as disease-modifying agents in PD.

## Figures and Tables

**Figure 1 molecules-26-03245-f001:**
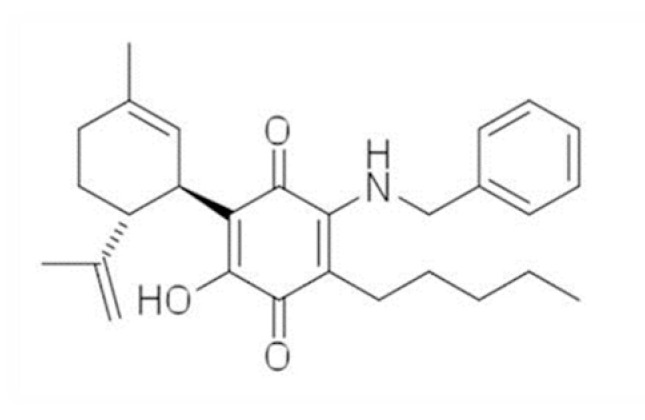
Chemical structure of VCE-004.8 (disclosed for the first time in [25]).

**Figure 2 molecules-26-03245-f002:**
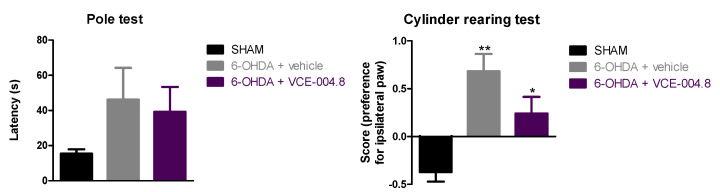
Response in the pole test and in the cylinder rearing test of control (sham) mice and unilaterally 6-OHDA-lesioned animals treated with VCE-004.8 (20 mg/kg) or vehicle (Maisine CC/corn oil) given orally. Treatments were daily and prolonged for 2 weeks. Data corresponded to 24 h after the last dose and were expressed as means ± SEM of more than 5 animals per group. They were analyzed by one-way ANOVA followed by the Bonferroni test (* *p* < 0.05, ** *p* < 0.01 versus sham).

**Figure 3 molecules-26-03245-f003:**
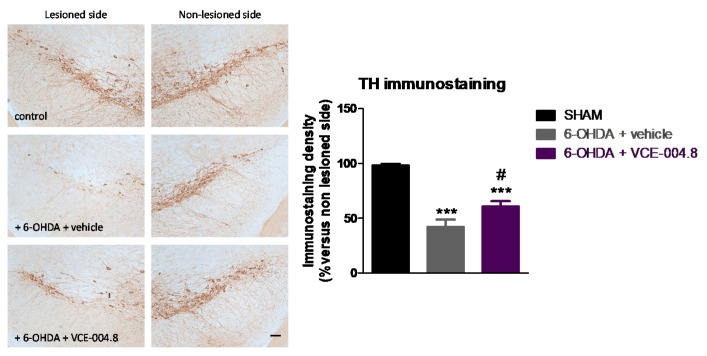
Intensity of the immunostaining for TH measured in a selected area of the substantia nigra pars compacta of control (sham) mice and unilaterally 6-OHDA-lesioned animals treated with VCE-004.8 (20 mg/kg) or vehicle (Maisine CC/corn oil) given orally. Treatments were daily and prolonged for 2 weeks. Data corresponded to percentage of the ipsilateral lesioned side versus contralateral non-lesioned side at 24 h after the last dose and were expressed as means ± SEM of more than 5 animals per group. They were analyzed by one-way ANOVA followed by the Bonferroni test (*** *p* < 0.005 versus sham; # *p* < 0.05 versus 6-OHDA + vehicle). Representative immunostaining images for each experimental group are shown in the left panels (scale bar = 100 µm).

**Figure 4 molecules-26-03245-f004:**
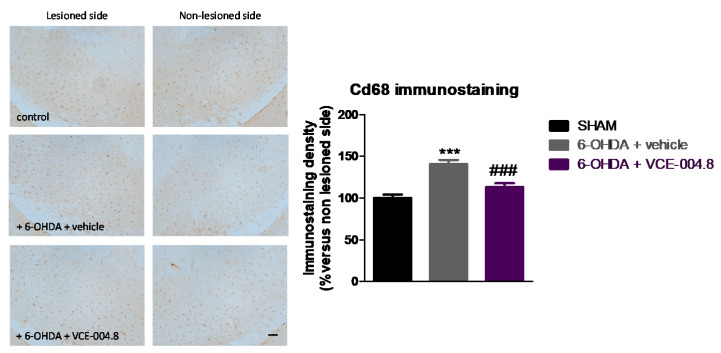
Intensity of the immunostaining for Cd68, a marker of microglia and infiltrated macrophages, measured in a selected area of the substantia nigra pars compacta of control (sham) mice and unilaterally 6-OHDA-lesioned animals treated with VCE-004.8 (20 mg/kg) or vehicle (Maisine CC/corn oil) given orally. Treatments were daily and prolonged for 2 weeks. Data corresponded to percentage of the ipsilateral lesioned side versus contralateral non-lesioned side at 24 h after the last dose and were expressed as means ± SEM of more than 5 animals per group. They were analyzed by one-way ANOVA followed by the Bonferroni test (*** *p* < 0.005 versus sham; ### *p* < 0.005 versus 6-OHDA + vehicle). Representative immunostaining images for each experimental group are shown in the left panels (scale bar = 100 µm).

**Figure 5 molecules-26-03245-f005:**
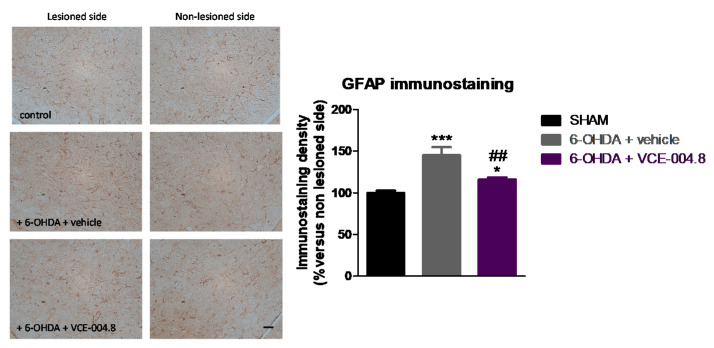
Intensity of the immunostaining for GFAP, a marker of astrocytes, measured in a selected area of the substantia nigra pars compacta of control (sham) mice and unilaterally 6-OHDA-lesioned animals treated with VCE-004.8 (20 mg/kg) or vehicle (Maisine CC/corn oil) given orally. Treatments were daily and prolonged for 2 weeks. Data corresponded to percentage of the ipsilateral lesioned side versus contralateral non-lesioned side at 24 h after the last dose and were expressed as means ± SEM of more than 5 animals per group. They were analyzed by one-way ANOVA followed by the Bonferroni test (* *p* < 0.05, *** *p* < 0.005 versus sham; ## *p* < 0.01 versus 6-OHDA + vehicle). Representative immunostaining images for each experimental group are shown in the left panels (scale bar = 50 µm).

**Figure 6 molecules-26-03245-f006:**
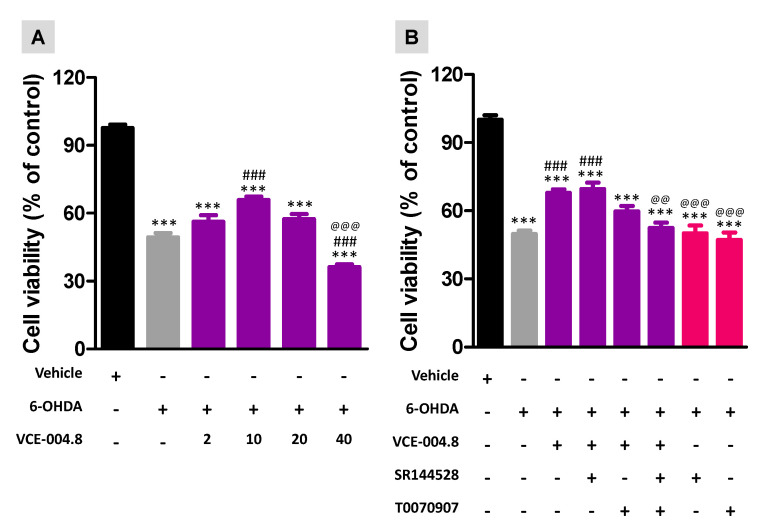
Cell viability measured with the MTT assay in cultured SH-SY5Y cells at 24 h to be treated with: (**A**): different concentrations of VCE-004.8 (2, 10, 20, and 40 µM) against 6-OHDA (200 µM), and (**B**): VCE-004.8 (10 µM) against 6-OHDA (200 µM), in the absence or the presence of T0070907 (10 µM), SR144528 (10 µM), or both. In all cases, a group with cells exposed to vehicles was also included to determine the 100% of cell viability. Values are means ± SEM of at least 4 independent experiments each performed in triplicate. Data were assessed by the one-way analysis of variance followed by the Bonferroni test (Both panels: *** *p* < 0.005 versus control cells; ### *p* < 0.005 versus cells exposed to 6-OHDA+vehicle; Panel A: @@@ *p* < 0.005 versus cells treated with the other VCE-004.8 concentrations; Panel B: @@ *p* < 0.01, @@@ *p* < 0.005 versus cells treated with VCE-004.8 in absence or presence or SR144528).

## Data Availability

Data supporting reported results may be supplied upon request by authors.

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
