# Peer review of "Neuroprotection with the Cannabidiol Quinone Derivative VCE-004.8 (EHP-101) against 6-Hydroxydopamine in Cell and Murine Models of Parkinson’s Disease"

_molecules, 2021, doi:10.3390/molecules26113245_

Round 1

Reviewer 1 Report

The article is well written, the conclusion were supported by the results and the methods were clearly explained. I do not have any  additional comments for the authors. In my opinion the article can be accepted in the present form.

Author Response

We thank the reviewer for the positive opinion on our manuscript

Reviewer 2 Report

Manuscript ID: molecules-1202971

          The article with title “Neuroprotection with the cannabidiol quinone derivative VCE-004.8 (EHP-101) against 6-hydroxydopamine in cell and murine models of Parkinson’s disease” by Javier Fernández-Ruiz et al investigated VCE-004.8 (formulated as EHP-101 for oral administration), for neuroprotective properties in experimental models of Parkinson’s disease (PD) in mice and in-vitro. The cannabidiol quinone derivative VCE-004.8 (EHP-101) with agonist activity at the cannabinoid receptor type-2 (CB2) receptor in addition to its activity at the PPAR-γ receptor was found beneficial for neuroprotection. The pharmacological effects of VCE-004.8 were confirmed on basis of the activity on the peroxisome proliferator-activated receptor-γ (PPAR-γ) rather than CB2 receptors by in-vitro experiments. This manuscript is valuable to the scientific community doing research in Parkinson’s disease.

General comments-

          Reference style is not as per Molecules journal format. Use of square brackets and number format is suggested. In certain cases the parenthesis give other information and seems confusing during reading.

          Structure of cannabidiol quinone to be indicated in a separate figure. In the abstract reference to VCE-003.2, is made, this structure should also be provided in the paper. In Line 73-105 all these compounds are referenced and no structures are provided. This is very essential for a journal like Molecules.

          Line 261-264; “Its pharmacodynamic profile (PPAR-γ and CB2 agonist) has been previously described (del Río et al., 262 2016) and its oral lipid formulation (EHP-101) is currently in clinical development (García-Martín et al., 2018).” Add “clinical development for bleomycin-induced skin and lung fibrosis”

          In section 4.3, it is mentioned administration was performed orally, yet the following sentence says injection, this is indeed confounding.

          Please check the references section for correctness and include the DOI format for all references.

Author Response

We thank the reviewer for the comments that will help to improve the manuscript. Following his/her comments, we have changed the reference style and numbering and included the DOI. We have also added a new figure (#1) with the chemical structure of VCE-004.8, and indicated the references where the structures of the other analogs were isclosed. We have  added the sentence in relation with its clinical development, and corrected the procedure for oral administration in section 4.3 (sorry for the mistake) 

Reviewer 3 Report

The submitted manuscript analyses the in vivo and in vitro effects of cannabidiol quinone derivative 2 VCE-004.8 (EHP-101) against 6-hydroxydopamine in murine models of Parkinson’s disease. In general the manuscript is well organized, data are well presented and presented results sustain the conclusion. I suggest the following few improvements:

  • What is the rationale for VCE-004.8 dose (20 mg/kg)? If this dose has been previously used, please add a reference. If not, dose response experiment is required.
  • The administration route of VCE-004.8 is not clear: if oral explain how it was administered, if injected specify the kind of injection (“VCE-004.8 (20 mg/kg) or vehicle were orally administered to 6-OHDA-lesioned 284 mice. Control mice (sham-operated) were also administered with the vehicle. The first injection, in all cases, was done approximately 16 hours after the lesion, and the treatment 286 was prolonged for two weeks (one injection per day, always at the same time)”. Lastly , what about the kind of vehicle?
  • Statistics in figure 5B is a bit confusing and not easy to follow in parallel to the main text. The presentation of statistically significant changes versus data presented in different panels does not help.
  • Move figure 1 from introduction into result section
  • References have to be numbered in the main text and not indicated as author first name et al. Reference list have to be changed accordingly (i.e. References must be numbered in order of appearance in the text)
  • Some abbreviations have not been defined at the first appearance in the main text
  • Slight revision of English text is required (i.e. cannabinoid receptor type-2 (CB2) receptor)

Author Response

Again, the comments from this reviewer will help to improve the manuscript, so many thanks for that. Following his/her recommendations, we have added a reference to justify the dose used for treatments, corrected the mistakes with the oral administration (as indicated for the reviewer #2), remarked the vehicle used (indicated only in the legends in the previous version), clarified the statistics in figure 5B (now 6B), moved the figure 1 (now figure 2) to the result section, and corrected the reference style (also indicated by the reviewer #2), the abbreviations not defined, and the slight English revision